# Chronic Myelomonocytic Leukemia Gold Jubilee

**Eric Solary** [1,2,*] and **Raphael Itzykson** [3,4]

1   Department of Hematology and INSERM U1287, Gustave Roussy, 94805 Villejuif, France
2   Faculté de Médecine, Université Paris-Saclay, 94270 Le Kremlin-Bicêtre, France
3   INSERM U944, Université de Paris, 75006 Paris, France; raphael.itzykson@aphp.fr
4   Service Hématologie Adultes, Hôpital Saint-Louis, AP-HP, 75010 Paris, France
*   Correspondence: eric.solary@gustaveroussy.fr

**Abstract:** Chronic myelomonocytic leukemia (CMML) was named 50 years ago to describe a myeloid malignancy whose onset is typically insidious. This disease is now classified by the World Health Organisation as a myelodysplastic syndrome (MDS)-myeloproliferative neoplasm (MPN) overlap disease. Observed mostly in ageing people, CMML is characterized by the expansion of monocytes and, in many cases, granulocytes. Abnormal repartition of circulating monocyte subsets, as identified by flow cytometry, facilitates disease recognition. CMML is driven by the accumulation, in the stem cell compartment, of somatic variants in epigenetic, splicing and signaling genes, leading to epigenetic reprogramming. Mature cells of the leukemic clone contribute to creating an inflammatory climate through the release of cytokines and chemokines. The suspected role of the bone marrow niche in driving CMML emergence and progression remains to be deciphered. The clinical expression of the disease is highly diverse. Time-dependent accumulation of symptoms eventually leads to patient death as a consequence of physical exhaustion, multiple cytopenias and acute leukemia transformation. Fifty years after its identification, CMML remains one of the most severe chronic myeloid malignancies, without disease-modifying therapy. The proliferative component of the disease that distinguishes CMML from severe MDS has been mostly neglected. This review summarizes the progresses made in disease understanding since its recognition and argues for more CMML-dedicated clinical trials.

**Keywords:** chronic myelomonocytic leukemia; myeloproliferative neoplasms; myelodysplastic syndromes

## 1. Introduction

Chronic myelomonocytic leukemia (CMML) is a clinically diverse, yet severe chronic myeloid malignancy that mostly affects elderly people. The expansion of pro-inflammatory, dysplastic and immunosuppressive monocytes and granulocytes is driven by the accumulation, in the hematopoietic stem cell compartment, of somatic variants that mostly involve epigenetic, splicing and signaling genes. In addition to generating increasing weakness and fatigue, the resulting insidious inflammatory climate creates deleterious cross-talks between mature and immature cells of the clone, probably affecting also bone marrow niche cells. This climate promotes clonal evolution at the expanse of wild-type hematopoiesis, eventually leading to patient death as a consequence of physical exhaustion, multiple cytopenias, and acute leukemia transformation. Fifty years after its naming, this relatively rare disease (0.4/100,000 inhabitants per year), which has long been neglected by the research community, remains incurable in the majority of patients as, beyond the rarely feasible allogeneic stem cell transplantation, there is still no really efficient, disease-modifying therapy. Behind the empirical search for effective new therapies, better understanding of CMML pathophysiology may suggest innovative strategies. The proliferative component of the disease, which frequently correlates with the acquisition of RAS pathway mutations and a severe outcome of the disease, deserves specific approaches. Such dedicated therapies will hardly be explored as long as CMML will be treated like severe myelodysplastic



syndromes (MDS) in clinical trials. Summarizing progresses made in disease understanding since its recognition as an individual entity, this review emphasizes CMML specificities and discusses how they could drive innovative therapeutic opportunities.

## 2. From Disease Identification to Current Definition

The designation CMML appeared in the early 1970s to describe a mixed monocytosis and granulocytosis with an insidious clinical onset and a relatively benign course, at least when compared to acute myelomonocytic leukemia [1,2]. Such a chronic leukemia, mostly observed in elderly patients, had been previously associated with refractory anemia and referred to under various terms including chronic monocytic leukemia [3], chronic erythromonocytic leukemia [4], subacute myelomonocytic leukemia [5,6] and preleukemia [7,8].

A first comprehensive analysis of CMML with mostly granulomonocytic features and only slight erythroid abnormalities was reported in 1975 [9]. All patients had a monocytosis above $0.8 \times 10^9$/L, correlating with an increased level of serum lysozyme, and blood films typically showed a mixed monocytosis and granulocytosis. Romanowsky staining identified so-called "paramyeloid" cells described as having "cytoplasmic and nuclear features intermediate between myelocytes and monocytes", which may be the cells described today as myeloid-derived suppressive cells (MDSCs). Chromosome analyses did not detect a Philadelphia chromosome but the clonal loss of chromosome Y was identified in one case, while another one demonstrated cytogenetic aberrations with secondary acute transformation. Importantly, most patients could survive several years, only a fraction of them undergoing transformation into rapidly fatal acute leukemia [9].

In 1982, the French–American–British (FAB) co-operative group included CMML as one of five newly defined myelodysplastic syndromes (MDS) [10] (Figure 1). The defining feature was an absolute monocytosis over $1 \times 10^9$/L, which was often associated with an increase in mature granulocytes, with or without evidence of dysgranulopoiesis. The percentage of blast cells in the peripheral blood had to be lower than 5%. The bone marrow showed a significant increase in promonocytes and a percentage of blast cells up to 20%.

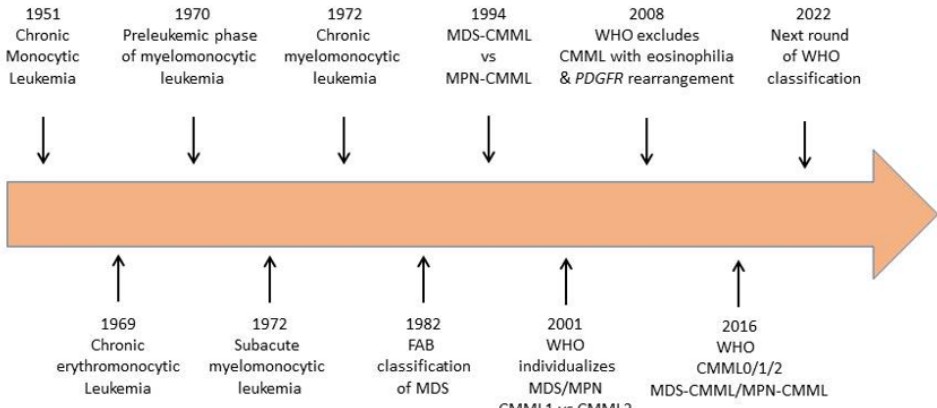

**Figure 1.** Evolution of disease name and classification over the last 70 years.

The FAB co-operative groups noticed that some CMML cases with organomegaly and high monocytic and granulocytic counts showed minimal dysplastic changes, nevertheless considered the disease as being closer to MDS than to myeloproliferative disorders as typical trilineage dyspoiesis could be observed in many cases [10]. Nevertheless, in 1994, the group divided CMML into a myeloproliferative (MP-CMML) and a myelodysplastic (MD-CMML) subtype using a cut point of WBC of $13 \times 10^9$/L [11]. The International Prognostic Scoring System (IPSS) group therefore elected to exclude CMML with a WBC of more than $12 \times 10^9$/L from its calculations [12].

When the World Health Organisation (WHO) reclassified hematological malignancies in 2001 [13], CMML was not more associated with MDS and became the most frequent

entity in a newly created, myelodysplastic/myeloproliferative neoplasm (MDS/MPN) overlap category. For the first time, cytogenetic and/or molecular examinations were included into disease definition to formally distinguish CMML from *BCR-ABL1* positive chronic myeloid leukemia. The WHO also separated CMML-1 (blasts plus promonocytes <5% in the PB, <10% in the bone marrow (BM) from CMML-2 (blasts plus promonocytes ≥5% in the PB, ≥10% in the BM), which proved to be clinically significant [14]. In contrast, the division between MD-CMML and MP-CMML temporarily disappeared. The 2008 revision of the WHO classification of myeloid neoplasms validated these changes but relocated some CMML with eosinophilia to the category "myeloid/lymphoid neoplasms with eosinophilia and *PDGFRB* rearrangement" [15].

According to the last iteration of the WHO classification that came out in 2016 [16], a diagnosis of CMML requires both the presence of persistent monocytosis $\geq 1 \times 10^9$/L and monocytes accounting for ≥10% of the white blood cell (WBC) differential count. The separation between MD-CMML and MP-CMML subtypes was re-introduced with the WBC cut point at $13 \times 10^9$/L. In addition, three blast-based groupings were proposed to improved prognostication, including CMML-0 (<2% blasts in PB and <5% blasts in BM), CMML-1 (2–4% in PB and/or 5–9% in BM), and CMML-2 (5–19% in PB, 10–19% in BM, and/or when any Auer rods are present). *BCR-ABL1* rearrangement should be excluded in all cases. When eosinophilia is present, *PDGFRA*, *PDGFRB*, *FGFR1* rearrangements or *PCM1-JAK2* fusions must be excluded [16] (Figure 1).

The WHO report emphasizes a precise morphologic evaluation to distinguish promonocytes from dysplastic monocytes and includes them into the blast cell count. These monocyte precursors display an abundant, finely granulated cytoplasm and a large, delicately folded nucleus with finely dispersed chromatin and a small or absent nucleolus. The WHO report also suggests integration of flow cytometry immunophenotyping and cytogenetic and molecular genetic testing in the disease characterization process.

Perspectives: The next iterations of CMML recognition and classification by the WHO may better incorporate recurrent genetic alterations and flow cytometry analysis of peripheral blood monocyte subset repartition. In a distant and still hypothetical future, disease recognition could include epigenetic markers and niche component alterations whose characterization is still in its infancy.

## 3. Flow Cytometry Improvement of CMML Recognition

Monocytes were described for the first time 140 years ago, two centuries after the initial description of peripheral blood cells by Antonie van Leeuwenhoek in 1674 [17] (Figure 2). Paul Ehrlich used acid and basic aniline coal tar dyes to classify white blood cells into mononucleated leukocytes, some of which he termed Übergangszelle (transitional cells) [18]. Ehrlich's Übergangszelle were subsequently coined as monocytes [19] and described as a morphologically homogenous blood cell population with a kidney shaped nucleus [20,21] (Figure 2A). At the end of the 20th century, the advent of flow cytometry identified some heterogeneity among circulating monocytes, based on the differential expression at their surface of CD14, a receptor for bacterial lipopolysaccharides, and CD16, which is the low-affinity receptor for immunoglobulin G (Fcγ-III receptor) [22]. The Nomenclature Committee of the International Union of Immunological Societies approved the subdivision of monocytes into three subsets [23], which was validated by gene expression profiling [24]. In healthy conditions, classical CD14+, CD16- monocytes represent roughly 85% of total human circulating monocytes, and express high level of the chemokine receptor CCR2, (CD192), which is the receptor of the cytokine MCP-1, but low levels of CX3CR1 receptor, which is the fractalkine receptor. They are distinct from CD14+, CD16+ intermediate monocytes and CD14low, CD16+ nonclassical monocytes, which express higher levels of CX3CR1 receptor (Figure 2B). In the last years, unbiased single-cell RNA sequencing approaches further broadened monocyte heterogeneity by identifying several subsets in human intermediate monocytes [25].

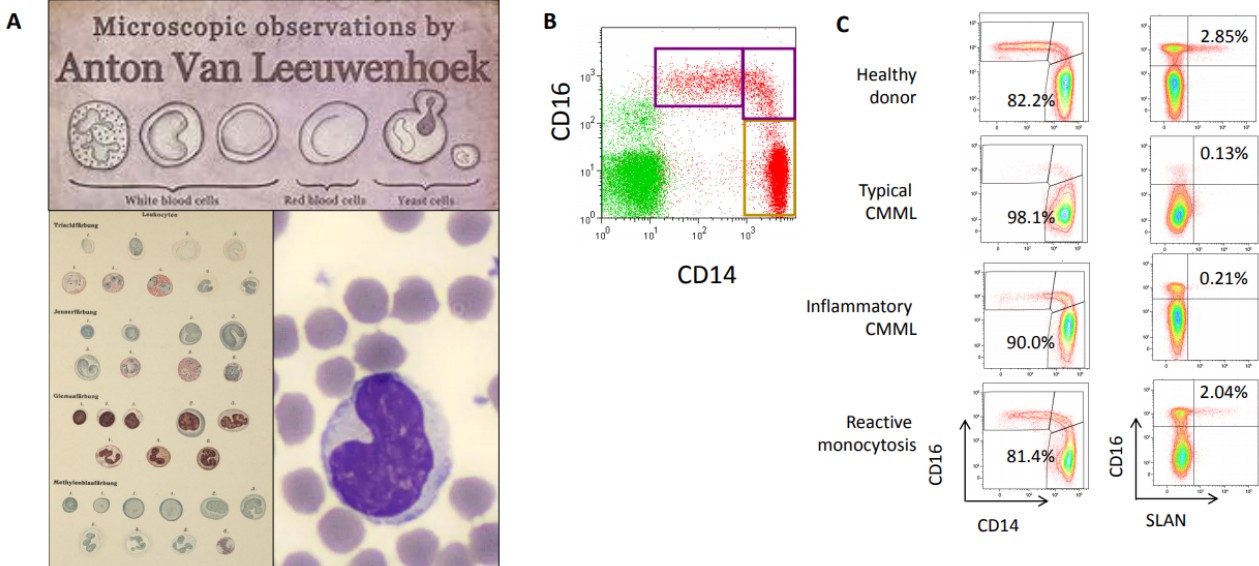

**Figure 2.** (**A**) From the first blood cell description by Antonie van Leeuwenhoek (1674, https://www.behance.net/gallery/4871853, accessed on 25 April 2021) and Ubergangszelle by Paul Ehrlich (O. Naegeli, Blutkrankheiten und Blutdiagnostik: Lehrbuch der morphologischen Hämatologie, https://www.archive.org/details/b31358457, accessed on 25 April 2021) to May–Grunwald–Giemsa staining (https://www.imaios.com/fr/imcases/training/565718/budorcas-taxicolor, accessed on 25 April 2021). (**B**) Flow cytometry analysis of peripheral blood monocyte subsets in healthy donor blood, showing classical CD14+, CD16−, intermediate CD14+, CD16+ and nonclassical CD14low, CD16+ monocyte subsets. (**C**) Monocyte subset repartition in a healthy donor, a typical CMML patient (classical monocytes >94% of total circulating monocytes), a CMML patient with associated inflammatory disease (classical monocytes <94% but decrease in Slan+ nonclassical monocytes) and a reactive monocytosis (Courtesy of Dorothee Selimoglu-Buet).

In patients with a CMML, a multiparameter flow cytometry assay quantifying the fraction of each monocyte subset among total peripheral blood monocytes typically shows an increase in CD14+, CD16− classical monocyte subset (also recognized as CCR2+, CX3CR1− monocytes) over 94% of total monocytes [26]. In contrast, this monocyte fraction is decreased in patients with a reactive monocytosis in which intermediate, and sometimes nonclassical, monocyte subsets accumulate (Figure 2C). This CMML-associated increase in CD14+, CD16− classical monocyte fraction is independent of age, WHO-defined disease subgroups and somatic mutations. Classical monocytes that accumulate in CMML patients show a distinct pattern of gene expression as compared to age-matched healthy donor monocytes, and sometimes demonstrate abnormal expression of CD56, CD115, and CD62L [27–29]. The 94% threshold was cross-validated by several independent groups Network [30–34], the assay was approved as a Clinical Laboratory Improvement Amendments-certified clinical test in the United States [30], and standardization is currently promoted by the European Leukemia [35]. In daily laboratory practice, the HematoFlow™ solution (Beckman-Coulter, Brea, CA, USA) that includes the cell surface marker CD16 can be used to screen for CMML suspicion in patients with a monocytosis before flow validation [36].

Flow cytometry detection of CMML can be challenged by the co-occurrence of an inflammatory disease that induces an increase in the intermediate subset of CD14+, CD16+ monocytes (Figure 2C), thereby decreasing the fraction of classical monocytes below the typical 94% threshold and erasing the CMML signature. In such a situation, detection of Slan, a carbohydrate modification of P-selectin glycoprotein ligand 1 (6-sulfo LacNac) that is typically expressed on a fraction of nonclassical monocytes, can be used to detect a CMML. A decrease in the Slan+ CD14low, CD16+ nonclassical monocyte fraction below 1.7% strongly argues for a CMML [37] (Figure 2C). Such a flow cytometry-defined inflammatory profile was shown recently to predict a poor outcome, independently of *ASXL1* gene mutation, high WBC count and cytopenias [38].

In the bone marrow, flow cytometry does not detect the abnormal monocyte subset repartition observed in the peripheral blood but identifies other populations of the leukemic clones such as plasmacytoid dendritic cells whose presence informs on disease outcome.

The median value of peripheral blood monocyte count increases with age, mathematical extrapolation indicates that this median value is ~$0.6 \times 10^9$/L by 100 years of age, the upper 95% confidence limit of the monocyte count being ~$0.8 \times 10^9$/L [39,40]. Thus, the absolute monocyte count can be technically increased below the threshold of a persistent monocytosis $\geq 1 \times 10^9$/L [41]. This may explain a fraction of so-called "oligomonocytic CMML" in which the absolute monocyte count is below $1 \times 10^9$/L but the monocyte fraction is $\geq 10$% [42–44]. Some of these patients have MDS that will never progress into CMML, the increased monocyte fraction being related to leukopenia and neutropenia. Others may evolve into genuine CMML [45]. We observed that 45% of those in which flow analysis of monocyte subset repartition detects an increase of classical monocyte fraction over 94% of total monocytes (a "CMML-like" phenotype) will demonstrate a monocytosis $\geq 1 \times 10^9$/L one year later [46].

Perspectives: Current flow cytometry approaches may be enriched with technological developments such as spectral flow and mass cytometry, adding to monocyte subset recognition the detection of immature granulocytes, dendritic cell subsets, and possibly other cell types, while incorporating intracellular signaling to further recognize and stratify the disease.

## 4. Incorporating Genetic Analyses in CMML Diagnosis

Cytogenetic analyses were initially reported mostly as single observations demonstrating that Philadelphia chromosome was not detected in CMML cells [2,5,9]. The absence of Philadelphia chromosome was subsequently introduced into the WHO classification as a CMML diagnosis criterion [13].

In the following years, large retrospective analyses identified chromosome abnormalities in leukemic cells of about 30% of CMML patients (Table 1), none being disease specific [47], and explored their prognostic significance [48,49]. By analyzing 414 patients from the Spanish MDS Registry, a CMML-specific cytogenetic risk classification was generated, distinguishing low risk (normal karyotype or loss of Y chromosome as a single anomaly), high risk (presence of Trisomy 8 or abnormalities of Chromosome 7, or complex karyotype), and intermediate risk (all other abnormalities). Multivariate analysis validated this CMML-specific cytogenetic risk stratification as an independent prognostic variable for overall survival [49], a stratification that was subsequently integrated to CMML-specific prognostic scoring systems (CPSS) [50,51]. A Mayo Clinic-French consortium suggested a slightly distinct stratification into low risk (normal karyotype, sole loss of Y and sole der (3q)), high risk (complex and monosomal karyotypes), and intermediate risk (all the others) that was efficient in predicting leukemic transformation [52,53].

Fifty years after disease identification, cytogenetic analyses, enforced by fluorescence in situ hybridization (FISH), remain mandatory in the diagnostic work-up of CMML [34]. Molecular analyses complete these cytogenetic investigations to eliminate *BCR-ABL1* fusion gene.

In 2008, CMML with eosinophilia were repositioned in a distinct WHO category in which fusion genes that are structurally and functionally analogous to *BCR-ABL1* involve the tyrosine kinase genes *PDGFRA*, *PDGFRB* and *FGFR1* or form the *PCM1-JAK2* fusion gene [54] (Table 1). Imatinib has become the standard of care for patients with *PDGFRA* and *PDGFRB* fusions as it generates long-term, deep molecular responses with uncommon secondary resistance [55]. In contrast, ruxolitinib generates mostly transient responses patient with JAK2 fusion gene [55,56]. As *PDGFRB* rearrangement was sometimes identified in CMML without eosinophilia, systematic profiling of these molecular features could integrate the routine diagnostic work-up of CMML [57].

**Table 1.** Main genetic alterations in CMML.

| Somatic Variants | Germline Predisposition * |
|---|---|
| **>10% of patients** | *ANKRD26* (ANKRD26-RT) |
| *ASXL1* | *ATG2B/GSKIP* |
| *CBL* | *DDX1* |
| *KRAS* | *ETV6* c.1160G > A (p.Arg369Gln) |
| *NRAS* | *GATA-2* (GATA-2 deficiency) |
| *RUNX1* | *RUNX1* (RUNX1-FTD) |
| *SRSF2* | |
| *TET2* | **Cytogenetic abnormalities ** ** |
| **5–10% of patients** | **Low risk** |
| *BCOR/BCORL* | Normal karyotype |
| *DNMT3A* | Loss of Y |
| *EZH2* | **High risk** |
| *JAK2* | Monosomy 7/deletion 7q |
| *PHF6* | Trisomy 7 |
| *SETBP1* | Complex karyotype |
| *SF3B1* | **Intermediate risk** |
| *U2AF1* | Deletion 20q |
| *ZRSR2* | Trisomy 21 |
| **<5% of patients** | Other: 3q-, 5q-, 12q-, 13q-, iso [17], +X etc . . . . |
| *ASXL2* | |
| *BRAF* | **CMML with MPN driver ** * ** |
| *CUX1* | *PDGFR1* rearrangement |
| *FLT3* | *PDGFRB* rearrangement |
| *IDH2* | *FGFR1* rearrangement |
| *IDH1* | *PCM1-JAK2* |
| *NF1* | |
| *PTPN11* | **SM-CMML ** ** ** |
| *TP53* | *KIT* |

Somatic mutations are grouped according to their frequency and classified by alphabetic order in each group. * ANKRD26-RT, ANKRD26-related thrombocytopenia or thrombocytopenia 2; RUNX1-FTD, familial platelet disorder with propensity to myeloid malignancies; *ETV6* c.1160G > A (p.Arg369Gln) is associated with familial thrombocytopenia 5; ** cytogenetic results are classified as low, intermediate and high risk according to CPSS. A normal karyotype is observed in 70% of cases; *** CMML with MPN driver are classified by the WHO as myeloid/lymphoid neoplasms with eosinophilia; **** SM, systemic mastocytosis.

In the last 15 years, rapid development of genomic techniques further inform on CMML associated molecular abnormalities. It appeared that, in many cases, disease appearance may be preceded by a clonal hematopoiesis of indeterminate potential (CHIP), also designated age-related clonal hematopoiesis (ARCH) [58,59]. Whole genome sequencing of circulating WBC identified two mutational signatures of ageing in each CMML patient [60], in accordance with the mean age of patients at diagnosis. Focusing on coding regions, each patient demonstrates a mean number of 14 somatic mutations, of which a mean number of three affect recurrently mutated genes [60–63]. These variants accumulate in hematopoietic stem and progenitor cells (HSPCs), the fraction of residual wild-type HSPCs is low, and the most mutated cells demonstrate a growth advantage that appears with cell differentiation [64].

Recurrently mutated genes are involved mostly in DNA methylation (*TET2, DNMT3A*), histone modifications (*ASXL1*), pre-mRNA splicing (SRSF2, *SF3B1, U2AF1, ZRSR2*) and signaling pathway (*NRAS, KRAS, CBL, JAK2*) [53,60–65] (Table 1). None of these mutations

are disease specific but the combination of mutations in *TET2* and *SRSF2* is typically observed in CMML [61]. Some of these somatic mutations are associated with a poor outcome and incorporated into prognostic scoring systems [51,61], signaling mutations are more commonly seen in proliferative CMML [66], while, contrary to MDS, *TP53* mutations are almost never detected in this disease.

The available evidence indicates that detection of somatic mutations in the peripheral blood is now part of the diagnostic process in patients with a peripheral blood monocytosis. Their pattern contributes to the recognition of CMML, even in the absence of definitive morphological criteria, and that of newly defined entities such as the previously mentioned oligomonocytic CMML [42–44,67,68]. In all these conditions, the variant allele frequency of somatic mutations is much higher than that observed in healthy subjects with CHIP/ARCH [58].

Some patients, usually younger, harbor an inherited predisposition to CMML (Table 1), although no single germline mutation has been exclusively associated with the disease. CMML can develop in the context of heterozygous inactivating mutation in the zinc-finger hematopoietic transcription factor *GATA2* (GATA-2 syndrome) [69], the transcription factor *RUNX1* (familial platelet disorder with propensity to myeloid malignancies) [70,71], the 5′ UTR of *ANKRD26* can result (ANKRD26-related thrombocytopenia) [72,73], the ETS family transcriptional repressor Ets variant 6 (*ETV6* c.1106G > A; p.Arg369Gln) [74], the DEAD-box RNA helicase gene *DDX41* [75,76], and the transcription factor TP53 [77]. Finally, in the French West Indies, germline duplication at chromosome locus 14q32 involving two genes, autophagy-related Protein 2 Homolog B (*ATG2B*) and GSK3β interacting protein (*GSKIP*), was identified in families from the French West Indies who developed a spectrum of myeloid malignancies including CMML [78].

Perspectives: The emergence of genetic (and epigenetic, see below) approaches at the single cell level may allow further deciphering CMML clonal diversity, within and between individual patients, and its evolution upon therapeutic pressure, and demonstrate the ability of new treatments to restore the fitness of persistent healthy cells.

## 5. Depicting the Role of Epigenetics in CMML Phenotype

The high prevalence of mutations in genes encoding epigenome-modifying enzymes such as *TET2* that is responsible for DNA demethylation [79–82] and *ASXL1* that is involved in histone-modifying complexes [83] may drive the aberrant epigenetic changes observed in CMML [84–88].

An aberrant DNA methylation was initially described at specific loci [89] and demethylation of the *p15(INK4)* cell cycle regulatory gene [90] was used as a pharmacodynamics marker of the efficacy of the nucleoside analogs decitabine (DAC) [91]. DNA hypermethylation was shown to increase with disease severity [92]. DAC and the other used hypomethylating drug, 5-azacytidine (AZA), restore a partially or totally balanced hematopoiesis in responding patients without significantly decreasing mutation allele burden in their circulating myeloid cells, suggesting a mostly epigenetic effect that correlates with DNA demethylation in patient hematopoietic cells [62]. A similar observation was reported in MDS in which AZA targets mostly granulomonocytic progenitors to alter the sub-clonal contribution to different lineages without eliminating founder clones [93,94].

Deregulated DNA methylation largely contributes to the clinical and biological expression of CMML. For example, the expression of the *TRIM33* gene, which encodes an E3 ubiquitin ligase (also known as transcription intermediary factor 1γ or TIF1γ) exerting a tumor suppressor function in hematopoietic cells, is commonly down-regulated by DNA hypermethylation in CMML. Deletion of this gene in myeloid cells is sufficient to mimic a CMML phenotype in mice, and *TRIM33* gene expression is restored to normal level in the cells of CMML patients who respond to hypomethylating agents [95]. Similarly, hypermethylation of a myeloid lineage specific regulatory sequence in *MIR150* gene was shown to account for the above-described deregulation of monocyte subset repartition in the peripheral blood [96]. miR-150 down-regulation in classical monocytes prevents

their transition to intermediate and nonclassical monocytes through up-regulating the TET3 (ten-eleven-translocation-3) protein in classical monocytes. This transition is restored in patients who respond to demethylating drugs [46,97]. Epigenetic down-regulation of another mi-RNA, miR-125a, is also reversed by HMA treatment [98]. Finally, the activation of transcripts containing endogenous retroviruses (ERVs) contributes to the therapeutic effect of AZA [99].

The baseline pattern of DNA hypermethylation, which varies among CMML patients, may predict the response to hypomethylating drugs [100] while genomic alterations have a limited predictive power [101]. The pattern of histone tail modifications at active promoters and enhancers in CMML cells, which may enhance chromatin mark modifications associated with stem and immune cell impairment with normal aging [102], could also influence the response to these drugs [99].

Perspectives: Future analysis of disease-associated changes in DNA methylation and histone marks, soon available at the single cell level, may depict the contribution of epigenetic abnormalities to clonal heterogeneity and therapeutic resistance, provide new criteria for treatment response, and guide new therapeutic approaches.

## 6. Dissecting the Role of the Inflammatory Climate

At the beginning of the 90s, CMML patient mononuclear cells were observed to spontaneously form granulo-monocyte colonies (CFU-GM) in semisolid cultures [103]. This effect was initially shown to be inhibited by antibodies targeting either interleukin-6 (IL-6) or granulocyte macrophage colony stimulating factor (GM-CSF), found at high concentrations in the supernatant of CMML, suggesting a paracrine effect [104]. The paracrine function of IL-6 was not consistently validated but addition of anti-GM-CSF antibodies reproducibly inhibited the spontaneous colony growth from CMML cells [105,106]. Further enforcing the importance of GM-CSF in CMML progression, the efficacy of xenotransplantation of patient cells in immunocompromised mice was improved by transgenic expression of human GM-CSF [107].

The ability of IL-4 to inhibit the spontaneous formation of colonies by CMML patient cells [105] has been controversial [108,109]. More consistently, IL-10 could inhibit the production of pro-inflammatory cytokines by monocytes [110,111] and, in most patients with a CMML, demonstrated a profound and dose-dependent inhibitory effect on autonomous in vitro growth of CMML cells [112]. This suppressive effect was reversed by the addition of exogenous GM-CSF and correlated with a substantial decrease in GM-CSF production by leukemic cells. A therapeutic effect of rhIL-10 was detected in a small pilot clinical trial, with no significant effect on WBC count but improvement of skin infiltration in a patient [113]. More recently, a decreased IL-10 plasma level correlated with poor overall survival of CMML patients, even when adjusted for *ASXL1* mutations and other prognostic features [114]. The negative prognostic value of circulating IL-10 now requires a solid, definitive validation.

GM-CSF hypersensitivity is the hallmark of juvenile myelomonocytic leukemia (JMML) cells. Spontaneous colony formation of JMML cells is abolished by prior depletion of monocytes, further supporting a paracrine mode of cellular proliferation [115]. Disease dependency on GM-CSF was validated by *Gmcsf* gene deletion in a mouse transgenic model of JMML generated by heterozygous mutation of *Nf1* tumor suppressor gene [116]. The dose-dependent inhibitory effect of IL-10 was also observed in the context of this pediatric disease [117].

Mouse model of chronic myeloid leukemia has identified feed-forward loops between mature and immature cells of the leukemic clone. Therapeutic targeting of cytokines involved was shown to inhibit disease installation or progression [118,119]. Based on these proof of principle experiments, targeting GMCF or its receptor could make sense in CMML and JMML.

The heterodimeric GM-CSF receptor associates with Janus kinase 2 (JAK2) whose subsequent phosphorylation initiates intracellular signaling events leading to a specific

evoked signal transducer and activator of transcription (STAT)-5 signature. This signature was used as a read-out of progenitor hypersensitivy to GM-CSF in JMML [117] and in most CMML [120]. Preclinical studies using either an antibody that prevents GM-CSF binding to its cognate receptor or chemical inhibitors of JAK2 supported a role for the GM-CSF/JAK2/STAT5 pathway in the proliferation of CMML myeloid progenitors and provided the rationale for testing JAK2 inhibitors in this disease. Ruxolitinib, a JAK1/2 inhibitor FDA approved for treatment of primary myelofibrosis (PMF), was shown to be a promising therapeutic in CMML [121]. Results form an extended evaluation of ruxolitinib activity in CMML patients may come out soon. More recently, a novel engineered immunoglobulin G1κ monoclonal antibody with high affinity for human GM-CSF (Lenzilumab, KB003) was introduced in CMML patients without major safety concerns [122].

Measurement of other cytokines, chemokines, and growth factors in CMML peripheral blood provided contrasted results. Initially, high levels of IL-6 and TNFα level were identified in the serum of a fraction of patients and elevated TNFα correlated with anemia [123,124]. More recently, annotation of inflammatory cytokines in plasma or serum in 213 patients classified CMML patients into three groups with distinct clinical and genetic features. A first cluster of diseases (32% of patients) was driven by a significant increase in M-CSF level, Cluster 2 (20%) by an increase in 17 cytokines including IL-6 and IL-8, and Cluster 3 (48%) by an increase in IL-2RA. This study also supported the poor prognostic value of a decreased IL-10 plasma level [114].

Single cell analyses detected increased expression of myeloid-lineage and cell cycle genes in CMML compared to healthy donor Lin$^-$CD34$^+$CD38$^-$ immature cells, together with a strong expression of interferon-regulatory factors in those collected from patients with the most advanced disease [125]. Among mature cells of the clone, the transcriptional signature of sorted CMML monocytes is highly proinflammatory when compared to age-matched healthy donor monocytes [126].

All these data suggest that the cytokine milieu, whose make-up involves immature and mature cells of the CMML clone as well as cells of the bone marrow niche, plays a role in CMML initiation and progression.

The lack of relationship between the inflammatory signature detected in monocytes and CpG island methylation pattern suggests complex regulatory mechanisms. One of these could involve *TET2* mutations that alter the non-catalytic functions of the protein [127]. In addition to promoting active DNA demethylation through iterative oxidation eventually leading to the replacement of 5mC by native cytosine, TET2 interacts with proteins that tether it to DNA [128–130] and exerts non-catalytic activities, e.g., by recruiting O-linked N-acetylglucosamine (O-GlcNAc) transferase (OGT) to gene promoters [131,132]. In mouse macrophages, a catalytically dead *Tet2* mutant represses *Il6* gene transcription, which contributes to the down-regulation an inflammatory reaction [133,134]. *TET2*-mutated clonal hematopoiesis was associated with an increased cardiovascular risk [135]. In mouse models, *Tet2* deletion associated cardiovascular risk was reduced by *Il6* gene deletion [136] or by inflammasome inhibitors [137]. Together, *TET2* mutations could have a differential impact on CMML outcome, depending on their effect on the expression level and non-catalytic activities of the protein that regulates the synthesis of multiple cytokines [138].

Altogether, the chronic inflammatory climate that increases with CMML severity involves still poorly understood interactions between immature and mature cells of the clone as well as between cells of the clone and their microenvironment, insidiously leading to physical exhaustion with disease progression.

Perspectives: An extended, well-controlled characterization of this inflammatory condition, i.e., by monitoring circulating cytokines and chemokines, will further inform on patient stratification and drive innovative therapeutic strategies to control the proliferative component of the disease.

## 7. Exploring the Role of the Micro-Environment

Chronic myeloid malignancies are increasingly seen as the rare outcome of a pervasive process of pre-neoplastic changes across a phenotypically normal hematopoietic tissue. Mutations accumulate with age in the stem and progenitor cell compartment, a mutant cell can expand into a clone without overt disease (CHIP), and rarely, such a clone eventually evolves into an overt malignancy such as a CMML. Accumulating experimental evidence points to a role of aging bone marrow niche in promoting clonal selection and evolution [139]. Malignant cells secondarily become independent of initially supporting stroma, i.e., disease become transplantable by injection of CD34+ hematopoietic cells only, yet diseased cells acquire an increasing potential to reprogram their environment to further support their expansion. The role and changes of bone marrow niche cells including immune cells, and the cytokines they release, have been scarcely explored in CMML patients. Bone marrow stroma cells from these patients behave differently from those of patients with an MDS, e.g., a decreased expression of *IL32* gene was detected in CMML stroma cells while this gene expression was increased in MDS stroma cells [140]. In addition, a procoagulant environment detected in CMML bone marrow niche was related to the exchange of tissue factor (TF) with clonal monocytes through extracellular vesicles [141]. Further investigation is mandatory to better understand how cells of the bone marrow niche contribute to disease emergence and progression, as depicted in other chronic myeloid malignancies such as primary myelofibrosis [142,143].

The role of immune cells in CMML is also largely unknown. Mature cells of the clone, including monocytes [144,145], immature, dysplastic granulocytes [146], and plasmacytoid dendritic cells [147], frequently exert immunosuppressive functions that promote either T-cell and NK cell death or T regulatory cell expansion. In addition, a fraction of MDS and CMML patients demonstrate PD-1 (Programmed Death 1) expression on stromal cells, while PD-L1 (Programmed Death Ligand 1) expression is increased in CD34-positive immature cells [148]. More recently, the expression of another immune checkpoint gene, LILRB4 (leukocyte immunoglobulin-like receptor subfamily B4), was reported to be increased in CMML compared to MDS and healthy control cells, correlating with CTLA-4 (cytotoxic T lymphocyte-associated antigen 4) gene expression [149]. While currently used, immune checkpoint blockers did poorly in myeloid malignancies, and further investigation will indicate if the immune context in which CMML emerges and progresses could offer new therapeutic opportunities.

Perspectives: There is a need for investigating bone marrow niche alterations in the context of CMML. With improved treatments, the question may appear whether eradication of leukemic cells will be sufficient to restore a physiological niche, as persistence of an altered bone marrow micro-environment could potentially promote the outgrowth of new clones or favor the evolution of a residual, ancestral one.

## 8. Generating Experimental Model Systems to Explore CMML

The lack of faithful experimental model of CMML remains a challenge to explore disease pathogenesis and identify innovative therapeutic strategies. None of the genetically engineered murine models generated for the three most commonly mutated genes (*Tet2* deletion, *Srsf2* P95H knock-in, *Asxl1* deletion) [89,150,151] or mimicking an epigenetically down-regulated gene such as *TIF1γ* [96] recapitulated all the key features of the disease, i.e., age-associated chronic elevation in monocytes and granulocytes, multilineage dysplasia, progenitor hypersensitivity to GM-CSF, and susceptibility to transformation to acute myeloid leukemia. Similarly, overexpression of the epigenetic regulator *Kdm6b* in mouse hematopoietic cells to mimic its overexpression in CMML patient cells generates a limited phenotype that appears upon stimulation with lipopolysaccharides, activating immunosuppressive genes such as the alarmin gene *S100A9* [152].

Closer to the human disease, concurrent *Tet2* loss and *Nras*$^{G12D}$ expression in hematopoietic cells induced a fully penetrant, lethal CMML-like phenotype that is sensitive to MAPK kinase (MEK) inhibition [153,154]. As in human disease, germline *Gata2* [155] or *Cbl* [156]

mutation could promote the development of a CMML-like disease in mice. Such a phenotype was observed in several other genetically engineered models as a step toward acute myeloid leukemia, including ablation of the BH3-only protein *Bid* [157], deletion of *Dok1* and *Dok2* adapter genes [158], and conditional ablation of *Tak1* (encoding TGF-β activated Kinase 1) [159] without identified relevance in the human disease.

Patient-derived xenograft (PDX) models of CMML can be reproducibly obtained by transplanting primary patient cells in immunocompromised mice. The robust engraftment of human bone marrow or peripheral blood mononuclear cells obtained in NSG mice is further enforced when these mice express human three human cytokines: stem cell factor (SCF), interleukin-3 (IL-3) and granulocyte/macrophage colony-stimulating factor (GM-CSF) (NSGS mice), likely through expression of GM-CSF [107,160]. In chronic phase, NSGS-engrafting leukemia-initiating stem cells reside in a CD45$^+$/CD34$^+$/CD38$^-$ fraction [161]. Even though these models allow functional analysis of clonal architecture [162] and preclinical evaluation of innovative therapeutic approaches [145], secondary transplantation only rarely succeed, and serial transplantation could be obtained only by lentiviral expression of a human oncogene in primary human CMML cells, as demonstrated using *MN1* gene [163].

Another approach to model the disease has been the generation of induced pluripotent stem cells (iPSCs) through reprogramming of CMML patient CD34-positive cells [164,165]. This approach captures a part of the disease genetic diversity and hematopoietic differentiation of the clones recapitulates the main features of the disease [165]. Some of these clones were shown to generate a humanized CMML mouse model via teratomas [164] and the model was used to dissect the individual contribution of recurrent genetic lesions such as mutations in the splicing factor *SRSF2* in leukemic cell behavior and to screen for drug sensitivities [166]. The relative refractoriness of malignant progenitors to reprogramming, e.g., due to their dyslastic nature that increases their sensitivity to apoptosis, and the difficulties in synchronizing the growth of the clones to perform comparative drug testing, are key limitations of these models [167].

Perspectives: We still miss the best experiment model to easily explore the pathogenic mechanisms involved in CMML and scree for therapeutic strategies that could be translated into clinics. Emerging approaches to generate immortalized cell lines from genetically modified animals and the emergence of 3D cultures are promising approaches to be developed and tested in this disease.

## 9. Depicting the Diversity of CMML Clinical Expression

Initial reports in the early 1970s described a relatively benign disease as compared to acute myelomonocytic leukemia [1,2]. Disease onset was reported as insidious in mostly elderly patients. Accordingly, diagnosis can be made in asymptomatic patients, based on a systematic blood survey identifying a peripheral blood monocytosis. This monocytosis is frequently neglected for months or years in patients who remain asymptomatic.

Nevertheless, initial reports also noticed the poor outcome of these patients as compared to other chronic myeloid malignancies [9]. Patients diversely develop constitutional symptoms (including weight loss, night sweats, and fever) and show consequences of hematopoietic insufficiency (fatigue, infections, and bleeding). About 25% of them demonstrate a splenomegaly [168]. Peripheral blood findings include leukocytosis with a monocyte or neutrophil predominance and sometimes the presence of immature and dysplastic myeloid cells. Macrocytic anemia and/or thrombocytopenia are commonly observed, neutropenia is less frequent and usually moderate. In rare cases, myelomonocytic infiltrates involve the skin, lymph nodes, and other extramedullary sites. Up to 25% of patients harbor a concurrent systemic inflammatory and autoimmune condition [169,170]. In some cases, dysregulated immunity was suspected to promote the emergence of CMML [171] but, in most cases, the chronic myelid malignancy promotes autoimmunity that correlates with mutations in the epigenetic regulators TET2 and IDH, the spliceosome component SRSF2, and T-cell lymphocyte imbalance [172,173].

The diversity of disease clinical and biological presentation is also observed at the pathological level. Bone marrow aspirate is consistently hypercellular but the blast cell fraction including promonocytes varies from 1 to 19%, cell dysplasia can be minimal, and ring sideroblasts, plasmacytoid dendritic cell [141] or mast cell infiltrates [174] and reticulin fibrosis [175] are detected in a minority of patients. Importantly, this heterogenous presentation does not fully correlate with mutation profiling [176].

Disease course is also highly variable with patients living for years with stable blood counts and few symptoms while others are highly symptomatic and succumb rapidly to their disease. All the patients share an increased risk of developing acute myeloid leukemia, which occurs in 15–20% of cases. Other patients die from progressive exhaustion, consequences of cytopenias and associated diseases including solid tumors [177].

**Perspectives**: With increased monitoring of otherwise healthy ageing people, CMML may be increasingly recognized at an early step, which may provide opportunities to better monitor the natural evolution of disease phenotype.

## 10. Looking for a Performant Prognostication Method

The poor outcome of some patients with a CMML was rapidly identified [178]. Survival of CMML patients ranges across a wide spectrum but their median overall survival (between two and three years) is shorter than that of any MPN and most MDS. Therefore, stratification factors are needed to guide personalized therapeutic choices such as allogeneic stem cell transplantation.

In 1994, the FAB divided CMML into dysplastic (MDS-CMML) and proliferative (MPN-CMML) sub-types, based on peripheral blood WBC with a cut point at $13 \times 10^9$/L [11]. The WHO subsequently replaced WBC by the blast cell count in the peripheral blood and the bone marrow [13]. In its last version, however, WBC was reintroduced in the WHO classification with the cutoff value initially proposed by the FAB. In addition, bone marrow and peripheral blood blast cell fractions (including promonocytes) distinguish three categories (CMML-0, CMML-1, CMML-2) with diverse outcomes [16]. Importantly, both elevated WBC and blast cell count correlate with the acquisition of gene mutations that activate the RAS signaling pathway, especially $NRAS^{G12D}$ [67].

In addition to the WHO-identified WBC and blast cell count, individual prognosis is related to other myeloproliferative features including splenomegaly, presence of circulating immature cells, and elevated lacticodehydrogenase (LDH) and to cytopenias (thrombocytopenia, anemia or red blood cell transfusion dependency). Starting in the 1980s, scoring systems incorporating demographics, clinical variables and peripheral and bone marrow findings were developed in MDS, including initially CMML as a disease subtype. Earlier models, such as Bournemouth [179], Spanish [180], and Düsseldorf [181] models, used clinical and peripheral blood counts. The Lille model [182] incorporated cytogenetics as a further refining stratification factor. In 1997, the unifying International Prognostic Scoring System (IPSS), which included cytopenias, bone marrow blasts percentage, and cytogenetics to define four risk categories, excluded CMML patients with WBC > $12 \times 10^9$/L [12]. In contrast, the MD Anderson prognostic scoring system, developed in 2008, which introduced age, performance status and transfusion dependency, was validated in a cohort of patients including proliferative CMML [183]. Four years later, the revised IPSS (IPSS-R), which divided MDS patients into five risk categories, again excluded CMML patients with proliferative disease [184].

A CMML-dedicated scoring system was first proposed in 2002, including peripheral blood absolute lymphocyte count, bone marrow blast count, hemoglobin level, and the presence of circulating immature myeloid cells [48,185]. Cytogenetics was included in the CMML-dedicated prognostic scoring systems (CPSS) in 2013 [50] and, the same year, molecular markers, starting with *ASXL1* gene mutations, were shown by the Groupe Francophone des Myelodysplasies to refine disease stratification [60,186]. Several other scoring systems dedicated to CMML were developed in the following years [52,187]. In 2015, retrospective analysis of a large international dataset of 1832 CMML patients treated across the

USA and Europe confirmed the independent prognostic relevance of nonsense/frameshift mutations in *ASXL1* gene, suggested that additional mutations such as those in *CBL* gene could indicate an adverse outcome, showed comparable performance of all the scoring systems, but also confirmed that a combination of clinical and molecular information may improve the accuracy of CMML prognostication [187]. Finally, an integrated scoring system, updating the previously described CPSS [50], introduced mutations in *RUNX1*, *NRAS*, *SETBP1*, and *ASXL1* in CMML prognostication [51].

In 2021, even though some prognostic factors have been validated in the context of treatment with DNA methyltransferase inhibitors [188], significant heterogeneity persists when using existing CMML scoring systems in clinical practice, guiding the timing of allogeneic stem cell transplantation, and assessing clinical trial eligibility.

Perspectives: More translational research characterizing the role of epigenetic marks, inflammatory marks, and bone marrow niche alterations in disease evolution, together with the application of machine learning to large international cohorts of homogeneously characterized patient, are needed to refine the models and provide a personalized prediction of overall survival and AML transformation.

## 11. Defining Appropriate Therapeutic Response Criteria

Consensus guidelines for the measurement of MDS response to therapy, which were initially defined in 2000 by the WHO [189], were updated in 2006 by the MDS International Working Group (MDS-IWG) [190]. In the following years, these guidelines were applied to clinical studies that included patients with CMML [191] and those conducted specifically for CMML [192–194].

In 2015, the MDS/MPN International Working Group (MDS/MPN IWG) proposed uniform criteria for MDS/MPN response to treatment, taking into account new parameters [195]. Complete response criteria included bone marrow blast cells <5% with normal cellularity, correction of myelofibrosis, improved blood cell counts (WBC $\leq 10 \times 10^9$/L; neutrophils $\geq 1.0 \times 10^9$/L; monocytes $\leq 1 \times 10^9$/L; neutrophil precursors $\leq$2%; hemoglobin $\geq 11$ g/dL; platelets $\geq 100 \times 10^9$/L and $\leq 450 \times 10^9$/L) with resolution of extramedullary disease and complete cytogenetic remission. Partial response was outlined as normalization of peripheral blood counts and splenomegaly, with bone marrow blast cells reduced by at least 50%. Finally, blood cell count and spleen size reduction with improved functional status, based on MPN-SAF scoring system developed in primary myelofibrosis [196], defined a provisional entity of "clinical benefit" [195]. These response criteria were validated in a retrospective cohort of CMML patients treated with hypomethylating agents [188].

Perspectives: These response criteria dedicated have to be evaluated in prospective studies, together with a comprehensive analysis of clinical symptoms and molecular abnormalities to confirm their contribution in defining robust short-term endpoints for future CMML clinical trials. These criteria could be subsequently refined with additional biomarkers, such as reduction of variant allele frequency in circulating myeloid cells and normalization of monocyte subset repartition and cytokine circulating levels.

## 12. Looking for Better Therapeutic Approaches

CMML patients have been poorly served by the research community so far, with a limited number of clinical trials dedicated to this disease since its initial description. Therefore, therapeutic management of CMML patients remains challenging [197].

A watch and wait approach is justified in lower-risk CMML patients exhibiting an indolent disease course without cytopenias, major proliferation or constitutional symptoms. In low-risk patients with anemia, supportive care includes red blood cell transfusions and, especially when serum endogenous EPO level is low, erythropoiesis-stimulating agents (ESA) [197,198]. Activin Receptor II ligand traps such as luspatercept and sotatercept could restore ineffective erythropoiesis in patients with low-risk MDS and CMML with acceptable safety profiles, suggesting that these drugs could represent a new paradigm for

anemia treatment in low-risk CMML [199,200]. In low risk CMML patients with isolated thrombocytopenia, the oral thrombopoietin receptor agonist eltrombopag is safe and could be considered [201].

Cytoreductive therapy is proposed to CMML patients with symptomatic leukocytosis or splenomegaly, in the absence of major cytopenias or excess of blast cells. The reference for cytoreductive therapy remains hydroxycarbamide, which was first approved in 1967. While it can effectively control WBC, hydroxycarbamide can also worsen a decrease in hemoglobin level and platelet count, introducing substantial difficulties in striking the optimal dosing balance. Gastrointestinal side effects are common but rarely require dose reduction or cessation of treatment. Some patients develop mouth or limb ulcers. In the first Phase III randomized clinical trial dedicated to CMML, hydroxycarbamide demonstrated superior outcomes compared to oral etoposide [202]. Various other drugs or combinations, including alpha-interferon [203] and retinoic acid [204], have been tested in the 1980s and 1990s, without significant response.

The only treatment modality with a curative intent is allogeneic stem cell transplantation [205]. Its potential remains limited by patient age at diagnosis and comorbidities, by significant rates of mortality caused by the procedure and by post-transplantation relapses. According to EHA/ELN guidelines, this treatment is recommended for higher-risk CMML patients below the age of 70 years, with an appropriately matched donor and no major contraindication to transplant [35]. It is more controverted in selected lower-risk patients with severe cytopenias or poor-risk somatic mutations [206]. In recent retrospective surveys, high-risk cytogenetics, mutations in *ASXL1* and/or *NRAS*, and increasing comorbidity index remained associated with worse survival after transplantation [207,208].

Hypomethylating agents, including 5-azacytidine (AZA) and decitabine (DAC), are commonly considered as the standard of care therapy for symptomatic CMML [209]. These drugs are usually well tolerated, with moderate cytopenias being the most common adverse event. Based on Phase 3 randomized trials dedicated to MDS with a low number of CMML included [210,211], both drugs have been approved by the FDA for CMML, while AZA is approved exclusively for dysplastic CMML-2 in Europe. The efficacy of these agents remains a matter of controversy [212]. Small, non-randomized clinical trials indicated a 40–50% overall response rate including 20% complete responses [192,213,214]. A large international retrospective study recently suggested that hypomethylating agents were the preferred therapy for patients with higher-risk or myeloproliferative CMML [215]. However, the only prospective randomized phase 3 trial comparing a hypomethylating agent (DAC) to hydrocarbamide as a frontline treatment of poor-prognosis CMML patients failed to demonstrate any advantage of DAC over hydroxycarbamide regarding event-free and overall survival [216]. Even when these drugs restore a balanced hematopoiesis, they hardly reduce the allele frequency of somatic genetic variant in leukemic cells, nor prevent genetic evolution of the clone. Therefore, disease relapse always occurs within a few months to two years after treatment initiation, without or with acute transformation [62] (Figure 3). Altogether, hypomethylating agents do not meaningfully alter the natural course of the disease [217].

In the last ten years, an increasing number of clinical trials recruiting CMML patients have been launched. Unfortunately, in most cases, CMML is still pooled with other myeloid malignancies, mostly MDS. Newly tested drugs include a new generation of hypomethylating agents with alternative modes of administration, as well as strategies combining HMAs with other compounds. In July 2020, the FDA approved the oral combination of decitabine (35 mg)/cedazuridine (100 mg) for adult patients with MDS and CMML, based on the results from two randomized crossover trials [218,219]. The well-tolerated AZA + lenalidomide combination may increase survival of CMML patients as compared to AZA alone [191] whereas the place of AZA + Venetoclax combination, which has recently emerged as a new standard of care for some newly diagnosed AML patients [220], has still to be defined in this specific disease as monocytes could generate some resistance to the combination [221]. Another molecule currently developed is pevonedistat, a small

molecule inhibitor of the DNA repair protein NEDD8, which demonstrated some clinical activity in combination with azacytidine [222]. Finally, the viral mimicry generated by hypomethylating drugs through gene demethylation was suggested to prime leukemic cells to immunotherapy, which remains to be validated in clinics [223,224].

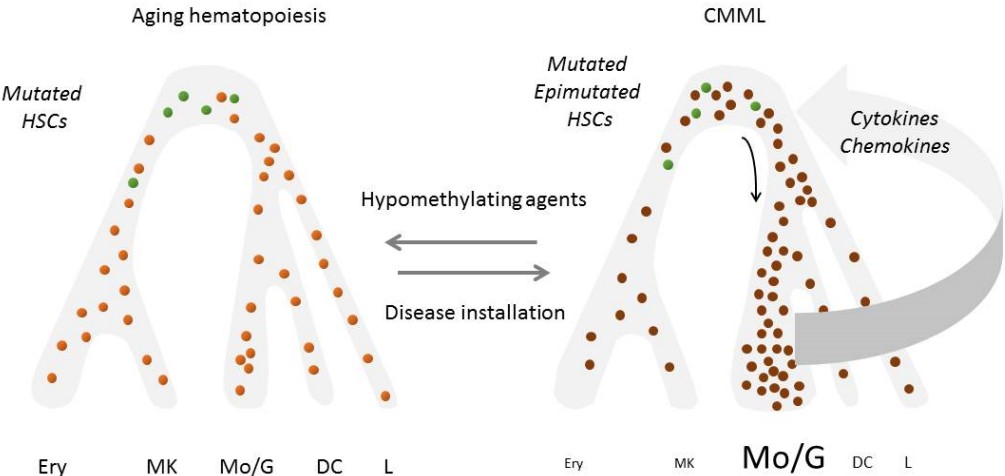

**Figure 3.** Evolution of hematopoiesis in CMML patients who respond to hypomethylating agents: demethylation of DNA restores a balanced hematopoiesis without decreasing gene mutation allele burden in peripheral blood cells (Reference 62). In green, residual wildtype cells; in light orange, mutated cells of the leukemic clone with demethylated DNA; in dark orange, mutated cells with hypermethylated DNA. HSCs, hematopoietic stem cells; Erythro, erythroid cells; Mkc, megakaryocytes; Lc, lymphocytes; Mc, monocytes; Gc, granulocytes; DCs, dendritic cells.

Importantly, the proliferative component of the disease deserves to be considered in the therapeutic strategy by targeting myeloid progenitor hypersensitivity to GM-CSF. The JAK1/2 inhibitor ruxolitinib induced objective responses in an early phase trial [193], and alternative approaches using pacritinib, a JAK2/FLT3 inhibitor, showed promising activity in patient derived xenografted mouse models [160]. Lenzilumab that targets human GM-CSF is an exciting therapeutic approach [122] and mavrilimumab, a GM-CSF receptor alpha directed monoclonal antibody [225] could be an alternative option.

Mutations in *RAS* genes and RAS-activating genes, which are common in proliferative CMML, have long been considered as prototypal undruggable oncogenic events. Nevertheless, new drugs have entered clinical trials [226] and targeting the RAS-MAPK pathway with MEK1/2 inhibitors [227] or a mitotic checkpoint kinase PLK1 inhibitor [228] or an immune based strategy [229] are currently tested approaches. CMML with a gain-of-function *CBL* mutation could be sensitive to the pharmacological inhibition of Lyn by dasatinib [230]. Finally, combined with an MCL1 inhibitor, a MEK inhibitor could target mature cells of the leukemic clone to disrupt feed forward loops between mature and immature cells of the clone [145].

Other potential therapeutic approaches include Glasdegib, an oral inhibitor of sonic hedgehog receptor smoothened in severe CMML [231], Imetelstat, a lipid-conjugated oligonucleotide targeting the RNA template of human telomerase reverse transcriptase [232], spliceosome inhibitors in CMML with splicing gene mutations [233,234], and the CD123-directed cytotoxin Tagraxofusp (formerly SL-401) in CMML with pDC accumulation [147,235,236].

Perspectives: while still insufficient, the number of CMML-dedicated clinical trials increases, which may give the patients a chance to receive in a disease-modifying treatment in the future.

## 13. Conclusions

Seventy years after its first description and fifty years after being named, CMML remains a severe malignancy. Most patients who are diagnoses with a CMML in 2021 will die of the disease evolution within a few years.

The WHO rightfully extirpated CMML from the MDS category, based on a typical proliferative component related to myeloid progenitor hypersensitivity to GM-CSF/STAT5 pathway activation. This separation may drive the use of therapeutic approaches targeting signaling pathways.

Most importantly, continued investment in research will refine disease associated changes in myeloid cell subset repartition, better detect the release of cytokines and chemokines, explore clonal evolution trajectories at the single cell level, decipher the poorly understood role of epigenetics, cell–cell interactions, and clonal cell environment in disease emergence and progression, and generate better experimental models of the disease. While improving disease identification and stratification, these investigations will also guide innovative therapeutic strategies.

**Author Contributions:** E.S. drafted a first version of the review, E.S. and R.I. generated the last version of the review. All authors have read and agreed to the published version of the manuscript.

**Funding:** This research received no external funding. E.S. group is member of the Equipe Labellisée Ligue Nationale Contre le Cancer (to Françoise Porteu) and supported by grants from Institut National du Cancer, Agence Nationale de la Recherche, Fondation ARC, Association Laurette Fugain, and Fondation AMGEN.

**Institutional Review Board Statement:** Not applicable.

**Informed Consent Statement:** Not applicable.

**Conflicts of Interest:** ES received a research grant from Servier Laboratories through the Molecular Medicine in Oncology Program supported by the Agence Nationale de la Recherche (Investissements d'Avenir).

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
