# Peer review of "Chronic Myelomonocytic Leukemia Gold Jubilee"

_hemato, doi:10.3390/hemato2030026_

Round 1
Reviewer 1 Report
This is well written review on CMML. Although not going into potentially relevant biological and diagnostics details e.g. the description of characteristic immunophenotypes used in diagnostics of malignant monocytes, this work is still welcome read for interested in the subject of this malignancy.
Author Response
We thank the reviewer for her/his positive comments. Actually, we dedicated a chapter to the characteristic immunophenotype that is used routinely to distinguih a CMML from a reactive monocytosis. This "Flow cytometry improvement of CMML recognition" chapter emphasizes several potential pitfalls in this approach and provides all the references to go into more details if needed. We added a new panel to revised Figure 2 to address this point.
Reviewer 2 Report
Herein Solary and Itzykson provided a comprehensive review on CMML. The work is divided in different paragraphs dealing with various aspects of the disease and each paragraph ends with few lines of future perspectives.
To improve the impact of this review and to help the reader follow the topic, I would suggest to: i) add a “table of content” at the beginning of the review listing all the paragraphs’ titles; ii) at the end, or in the “Conclusions” section, include a list extrapolated from the text of all the things researchers should focus on to study/better characterize/cure the CMML [eg, a) improve current flow cytometry diagnostic for CMML; b) better define the prognostic value of IL-10; c) schedule experiments to explore marrow niche, immune cell and cytokine release in CMML; d) define the role of immune cells in CMML; e)…. Etc.. etc…].
In legend to Figure 1, please correct the year.
As for Figure 2, I would separate the different figure in a) for “first blood description”, b) for “May-Grunwald/Giemsa staining” and c) for “flow cytometry”. I would also add a fourth figure (d) representing genetic lesions in CMML. Please, cancel (Figure 2) from line 83.
Please, check for typos throughout the text.
Author Response
We thank the reviewer for these suggestions.
Introducing a table of content does not fit with editor’s recommendation. Nevertheless, swith the editor’s authorization, such a table of contents could be
- Introduction
- From disease identification to current definition
- Flow cytometry improvement of CMML recognition
- Incorporating genetic analyses in CMML diagnosis
- Depicting the role of epigenetics in CMML phenotype
- Dissecting the role of the inflammatory climate
- Exploring the role of the micro-environment
- Generating experimental model systems to explore CMML
- Depicting the diversity of CMML clinical expression
- Looking for a performant prognostication method
- Defining appropriate therapeutic response criteria
- Looking for better therapeutic approaches
- Conclusions
- References
As an alternative to this table of content and following another comment by reviewer 3, we propose a revised abstract that describes the topics addressed in this review in a more systematic manner.
We also modified the conclusions following the reviewer’s recommendation. New conclusions are as follows:
Fifty years after being named, CMML remains a severe malignancy. Most patients who are diagnoses with a CMML in 2021 will die of the disease evolution within a few years.
The WHO rightfully extirpated CMML from the MDS category, based on a typical proliferative component related to myeloid progenitor hypersensitivity to GM-CSF/ STAT5 pathway activation. This separation may increasingly drive the use of therapeutic approaches targeting signaling pathways.
Most importantly, continued investment in research will refine disease associated changes in myeloid cell subset repartition, better detect disease-associated release of cytokines and chemokines, explore clonal evolution trajectories at the single cell level, decipher the poorly understood role of epigenetics, cell-cell interactions, and clonal cell environement in disease emergence and progression, and generate better experimental models of the disease. While improving disease identification and stratification, these investigations will also guide innovative therapeutic approaches.
Unfortunately, we had to delete Figure 1. We asked to the editor (Wiley) of the first manuscript describing CMML to get access to the first page as a historical document for this “gold jubilee”’s manuscript. We did not receive any answer to our messages and the 1972 issues of British Journal of Haematology was not found in our library nor in Hospital Saint-Louis library.
As for Figure 2, which became figure 1, we separated cytology and flow cytometry. We provide additional flow images as panel 3 to better illustrate the characteristic phenotype associated to the disease. Disease-associated genetic events (including somatic mutations, germline predisposition, cytogenetic abnormalities as for CPSS, and systemic mastocytosis mutation) are provided in an independent table 1.
Figure 2 was removed from line 83, thank you for mentioning this error.
We did our best to correct the typos along the manuscript
Reviewer 3 Report
In this review, the authors provide an overview of “chronic myelomonocytic leukemia” (CMML) and summarize the progresses made in understanding this disease in recent years. They also elaborate on the present state of clinical trials and therapeutic options for CMML as well as an outlook on future research perspectives.
The present manuscript is comprehensively structured and well phrased. It gives a great general overview of the progresses made since the first classification of CMML and is adequately evidenced by current literature. The included figures are reasonably placed.
Overall, although this review is informative, very interesting and relatively pleasant to read in most parts, it would benefit from revision by an english native speaker.
Minor:
- Syntax mistake in line 463: double use of word “human”
- Line 441: erroneous grammar
- In the abstract, specifically lines 23-24, it does not become clear what the content of the review will be. I suggest being more precise and roughly outlining the content here.
- Some sentences are overly long, e.g. ll33 – 38.
Author Response
We thank the reviewer for his/her positive comments and suggestions to improve the manuscript
Minor:
- Syntax mistake in line 463: double use of word “human”
This error has been corrected
- Line 441: erroneous grammar
This error has been corrected
- In the abstract, specifically lines 23-24, it does not become clear what the content of the review will be. I suggest being more precise and roughly outlining the content here.
Based on comments by reviewer 2 and 3, the abstract was totally modified in order to better describe the content of the review
Chronic myelomonocytic leukemia (CMML) was named 50 years ago to describe a myeloid malignancy whose onset is typically insidious. This disease is now classified by the World Health Organisation as a myelodysplastic syndrome (MDS)-myeloproliferative neoplasm (MPN) overlap disease. Observed mostly in ageing people, CMML is characterized by the expansion of monocytes and, in many cases, granulocytes. Abnormal repartition of circulating monocyte subsets, as identified by flow cytometry, facilitates disease recognition. CMML is driven by the accumulation, in the stem cell compartment, of somatic variants in epigenetic, splicing and signaling genes, leading to epigenetic reprogramming. Mature cells of the leukemic clone contribute to creating an inflammatory climate through the release of cytokines and chemokines. The suspected role of the bone marrow niche in driving CMML emergence and progression remains to be deciphered. The clinical expression of the disease is highly diverse. Time-dependent accumulation of symptoms eventually leads to patient death as a consequence of p
- Some sentences are overly long, e.g. ll33 – 38.
We have modified some of the sentences to make them shorter
Reviewer 4 Report
The article is a cross-sectional analysis of pathogenesis, diagnosis and treatment of CMML, taking into account the development perspective in the near future. The article is valuable and well written.
Comments:
- Please changing the title to "Chronic Myelomonocytic Leukemia Gold Jubilee" as the term disease appeared 50 years ago.
- Please consider increasing the quality of Fig 4. In addition, in the description of this figure, there should be explanations of the abbreviations.
Author Response
We thank the reviewer for her/his positive comments and useful suggestions
The title has been modified following the provided recommendation
Figure 4 (now figure 3) was improved and we provide a list that explains the abbreviations in the figure legend